# The Patient Journey of Schizophrenia in Mental Health Services: Results from a Co-Designed Survey by Clinicians, Expert Patients and Caregivers

**DOI:** 10.3390/brainsci13050822

**Published:** 2023-05-19

**Authors:** Mauro Emilio Percudani, Rosaria Iardino, Matteo Porcellana, Jacopo Lisoni, Luisa Brogonzoli, Stefano Barlati, Antonio Vita

**Affiliations:** 1Department of Mental Health and Addiction Services, Niguarda Hospital, 20162 Milan, Italy; 2Fondazione The Bridge, 20156 Milan, Italy; 3Department of Mental Health and Addiction Services, ASST Spedali Civili of Brescia, 25123 Brescia, Italy; 4Research Department, Fondazione The Bridge, 20156 Milan, Italy; 5Department of Clinical and Experimental Sciences, University of Brescia, 25123 Brescia, Italy

**Keywords:** early detection, expert patient, mental health, patient journey, peer support, prevention, recovery, schizophrenia, stakeholder engagement, treatment

## Abstract

Background: The Patient Journey Project aims to collect real-world experiences on schizophrenia management in clinical practice throughout all the phases of the disorder, highlighting virtuous paths, challenges and unmet needs. Methods: A 60-item survey was co-designed with all the stakeholders (clinicians, expert patients and caregivers) involved in the patient’s journey, focusing on three areas: *early detection and management*, *acute phase management and long-term management/continuity of care*. For each statement, the respondents expressed their consensus on the *importance* and the *degree of implementation* in clinical practice. The respondents included heads of the Mental Health Services (MHSs) in the Lombardy region, Italy. Results: For *early diagnosis and management*, a strong consensus was found; however, the implementation degree was moderate-to-good. For *acute phase management*, a strong consensus and a good level of implementation were found. For *long-term management/continuity of care*, a strong consensus was found, but the implementation level was slightly above the cut-off, with 44.4% of the statements being rated as only moderately implemented. Overall, the survey showed a strong consensus and a good level of implementation. Conclusions: The survey offered an updated evaluation of the priority intervention areas for MHSs and highlighted the current limitations. Particularly, early phases and chronicity management should be further implemented to improve the patient journey of schizophrenia patients.

## 1. Introduction

Schizophrenia is a severe mental disorder characterized by a debilitating progression in most cases. Despite its etiology not being completely understood, schizophrenia seems to result from a complex interaction of biological, genetic and environmental factors [1]. According to the DSM-5 criteria [2], clinical features of schizophrenia comprise positive (delusions and hallucinations), negative (anhedonia, avolition, alogia, asociality, blunted affect), and disorganization (formal thought disorder, disorganized behavior) symptoms [1,3,4]. Moreover, cognitive impairment represents a further core feature [1]: in particular, impaired neuro-cognitive functioning exhibited the strongest association with poor psychosocial functioning [5,6].

Despite a low prevalence of ~1%, about 26.3 million people are currently living with schizophrenia [7]. Indeed, schizophrenia is ranked among the leading causes of disability worldwide [8,9]. Slightly more common in men [1], schizophrenia seems to present with a highly variable yet definitively chronic course in almost 60% of cases [10] and low recovery rates [11], negatively affecting subjective well-being, quality of life and psychosocial functioning [1,3]. 

Furthermore, people living with schizophrenia present with a low life expectancy due to the occurrence of other psychiatric disorders (i.e., depression, anxiety, substance misuse) [12] and somatic comorbidities (i.e., metabolic syndrome, diabetes, cardiovascular, respiratory and infectious diseases) [13,14]. Therefore, achieving common psychosocial milestones is uncommon for people living with schizophrenia [15]. This places great socioeconomic burdens on health care systems, mainly due to the indirect costs (i.e., unemployment, social support and hospitalization during crises) [16]. Indeed, while the annual amount ranges from USD 94 million to USD 102 billion [17], great costs also indirectly result from the increased family burdens and the reduced quality of life for relatives and caregivers [18]. 

Moreover, great burdens are also derived from the so-called dual diagnosis condition, which is when patients affected by severe psychiatric disorders suffer from concomitant substance use disorders (SUD). Indeed, data from an Italian study showed that the management of comorbid SUD in patients with schizophrenia is increasingly complex, highlighting an urgent need to optimize the management of this difficult-to-treat condition by considering several factors (i.e., treatment efficacy, tolerability, metabolic effect sides) and, according to a multidisciplinary approach, throughout all the phases of both disorders [19].

It, therefore, seems obvious that the management of patients with schizophrenia is complicated by the various and multifaceted elements that have to deal not only with the course of the disease but also with the different phases of the patient’s life. To address these challenges, mental health care systems (MHS) and providers have to offer updated treatment plans to take care of the individual suffering from schizophrenia throughout all the phases of what may amount to an ideal patient’s journey, with the aim to increase the functional outcomes and reduce the risk of chronicity [20], particularly focusing on the clinical features (i.e., negative symptoms and cognitive impairment) linked to poor functional outcomes [21]. Indeed, a previous Delphi study explored the consensus of Italian experts, psychiatrists and trainees in psychiatry and showed high consensus on several components of schizophrenia care, including early recognition, personalization, integration of care, assessment standardization and the management of somatic and psychiatric comorbidities [20].

Thus, the patient’s journey has to focus on particular moments throughout all the phases of the disorder, which are the *early detection*, the *acute phase management*, and the *long-term management/continuity of care*, with an eye open to the personalization of this path. However, as we believe that the patient’s journey should not be built based solely on a clinician-oriented perspective, this ideal path should consider the opinions and the needs of all the stakeholders involved in the care of patients with schizophrenia.

Therefore, the purpose of this Patient Journey Project is to perform a survey to share evidence-based information and real-world experiences on schizophrenia management, with the specific aim of involving all the stakeholders (i.e., clinicians, expert patients, caregivers and family members) engaged in the planning of the ideal path of care for patients with schizophrenia. In the manuscript, the ideal and virtuous pathways of care, as well as the challenges, barriers and difficulties in their implementation, will be taken into account. Taking care of the schizophrenic patient implies, on the one hand, the necessity for highly specialized care and, on the other hand, the need for multidisciplinary skills distributed throughout the community healthcare setting. The idea to plan a patient’s journey in schizophrenia through the organization of a codified path of care and management, giving specific assistance and improving adherence to pharmacological and psychosocial treatments, may represent a response to the complexity of schizophrenia, with the ultimate goal to achieve recovery in this population. 

Thus, we first provide an up-to-date scenario of the current knowledge and existing matters on three main themes of *early detection and its management*, *acute phase management* and *long-term management/continuity of care*. Then, we will discuss the results of the survey to examine the unresolved needs outlined by the stakeholders and identify possible gaps and areas to be further implemented throughout the three phases of the patient’s journey while also proposing solutions and recommendations.

### 1.1. Early Detection and Management

Regarding early management, MHSs should implement structural plans to improve treatment outcomes and reduce the worst consequences of a full-blown psychotic disorder [22]. In order to foster an adequate treatment plan for adolescent help-seekers with mental disorders, several factors should be considered, including the family milieu, environmental (i.e., illicit substances abuse) or traumatic exposures, the duration of untreated psychosis (DUP), the onset/course of the disorder and its clinical manifestations (i.e., negative symptoms, social and non-social cognitive impairments), social functioning and quality of life, concomitant psychiatric/physical comorbidities, as well as resilience and internalized stigma levels [23]. 

As schizophrenia generally occurs in late adolescence/early adulthood, this life stage represents an essential period towards which the efforts of MHSs must be directed to identify the early signs of a forthcoming disorder and to prevent further clinical deterioration [24]. Indeed, according to the neurodevelopmental hypothesis of schizophrenia [25], adolescence represents an essential moment for brain maturation [26], during which several stressing factors, both biological (disturbed pruning, altered myelinization, exposure to cannabis or other illicit drugs and genetic load) and psychosocial (increased academic demands and responsibilities, social stress and social deprivation), could negatively impact normal brain development significantly earlier than the illness onset [27]. Thus, the Ultra-High Risk (UHR) model [28,29,30], including the *Attenuated Psychotic Symptoms (APS)*, *Brief Limited and Intermittent Psychotic Symptoms BLIPS*, and the *Genetic Risk and Deterioration factor* definitions, was developed to classify *at-risk* individuals in a prodromal phase of a psychotic disorder, with the aim to delay and prevent the onset of a full disorder, further reducing the impact of unfavorable factors (e.g., the duration of untreated illness (DUI) and the DUP) that could deteriorate psychosocial functioning and quality of life [28]. However, the recognition of subtle psychotic symptoms in *at-risk* subjects is frequently made by family and teachers rather than healthcare professionals. In this context, MHSs should also conceive preventative measures to identify *at-risk* adolescents through public awareness campaigns, social media, public events and community works to educate those who are part of the individual social context (e.g., academic milieu, general practitioners and local public health authorities) on the early manifestations of the disorder. Thus, assuming that the timing of treatment of the first episode of psychosis (FEP) is a crucial factor in determining the prognosis, several international programs [31,32,33,34,35], including Italian ones [36,37], have been developed to target both *at-risk* help-seekers or FEP patients, demonstrating the effectiveness of these interventions at improving symptom severity, retention rates and treatment adherence as well as the quality of life and psychosocial functioning [38]. The crucial point is to deliver a tailored intervention addressing the patient’s needs through a team-based multidisciplinary approach that involves psychiatrists, psychologists, nurses, social care workers, general practitioners (GPs) and families and comprises providing pharmacological, psychological, psychoeducation, psychosocial interventions, rehabilitation, family therapy and supported employment interventions [39]. Moreover, given the feasibility of receiving psychosocial and/or pharmacological treatments aimed at reducing the perceived distress at the early stages, it seems that early detection/intervention services could result in considerable cost-savings for national health systems, reducing hospitalization rates and improving employment outcomes [40]. 

Although the transition rate to full-blown psychosis is relatively low (20–35% over 2 years), clinicians and national legislators are called upon to appropriately respond to the adolescent population transitioning from the Child and Adolescent Mental Health Services (CAMHS) to Adult Mental Health Services (AMHS) [41], in order to reduce the personal, familiar and societal costs and burdens. However, several barriers prevent optimal collaboration between these services, hindering a real understanding of personal and family needs as well as spreading stigma towards AMHS [42,43]. In Italy, the National Action Plan for Mental Health [44] provided recommendations to develop innovative plans, creating multidisciplinary teams that involve CAMHS and AMHS together with families, educational facilities and the environmental context to share all information and recommendations on the clinical course of the disorder at an individual level. Conversely, bridging the gap in the transition phase and providing continuity of care, the *Italian Partnership for Psychosis Prevention (ITAPP)* project included five national Clinical High Risk for Psychosis (CHR-P) academic centers aimed at serving both adolescents and young adults with multidisciplinary and integrated interventions [45]. 

In this scenario, a central point of early management should be to promote simplified access to MHSs, promoting close connections between the AMHS, CAMHS, GPs and other parties operating in the social and health network of the patient, providing multi-professional interventions within the family to address patient’s physical and psychosocial needs through psychoeducational, psychotherapeutic and rehabilitative interventions [46]. 

Regarding therapeutic management, international guidelines [24,47,48] advise professionals to provide interventions addressing all the needs of the patient and their family, especially in the case of *at-risk* conditions (e.g., psychological support; cognitive behavioral therapy (CBT) and family-oriented interventions; continuous physical health assessments promoting well-being and healthy diet, including physical activity, smoking and psychoactive substance misuse cessation; education and employment support), where pharmacological intervention is highly recommended during an acute crisis, particularly with FEP [46]. 

### 1.2. Acute Phase Management 

While long-term antipsychotic treatments prevent the relapse of the disorder both after a first episode and in the case of a chronic course [49,50], delaying time to hospitalization, especially in the early phases of the disorder [51], the management of the acute phase of schizophrenia should consider an adequate antipsychotic therapy [46] and, if recommended, a hospital admission, thus avoiding involuntary hospitalization where possible [46,52]. However, the patient’s journey usually begins with an acute crisis that, especially in the case of FEP, is generally followed by immediate hospitalization. Thus, as involvement in early diagnosis programs is usually difficult, the implementation of outpatient services should be pursued, preventing acute crises and involuntary treatments [46].

When hospitalization occurs, given the recognized efficacy of both first- (FGAs) and second-generation antipsychotics (SGAs) during the acute phase [53], clinicians should provide personalized treatments after careful assessment of the ratio between the benefits derived from symptom control and the risk of adverse events due to the chosen drug [20,54]. Moreover, the international guidelines advise a careful evaluation of the patient’s clinical presentation (i.e., symptom severity, suicidal risk, agitation/aggressiveness and psychiatric comorbidities, including substance abuse disorders) and of somatic comorbidities to maximize treatment adherence, tolerability and efficacy [55,56]. A feasible option is to receive long-acting injectable (LAI) antipsychotic treatments and clozapine for treatment resistance, as they are associated with higher reductions in hospitalization rates when compared to oral antipsychotics [57] and a reduced risk of relapse and recurrence [58]. However, as prolonged inpatient treatment of the general psychiatric and forensic populations and the use of coercive treatments (i.e., forced physical seclusion, restraint, forced medication treatments) [59,60,61,62] currently represent unethical methods [46], national MHSs are claimed to have developed effective intervention programs that might integrate the family and psychoeducational interventions during the acute phases, improving therapeutic alliance and treatment adherence [63], reducing those factors associated with longer hospitalization stays (e.g., the DUP) [64] while always ensuring the continuity of care with community setting services [46]. 

### 1.3. Long-Term Management/Continuity of Care 

As the main goals of this phase are to prevent relapses and recurrences and to maintain remission and achieve recovery, in November 2014, the Italian Ministry of Health provided a clear definition of the long-term management phase, firmly recommending the implementation of the integration and continuity of care services, endorsing the application of multi-professional community-based interventions through the definition of the so-called *Individual Treatment Plan* [46,65]. If the initial suggestion is to avoid drug discontinuation through LAI treatments [66,67] and to carefully manage substance abuse disorders [65], then further recommendations should concern the application of non-pharmacological *evidence-based* interventions, such as psychoeducation for patients and their families [68], problem-oriented therapy, CBT for resistant-positive symptoms [69], cognitive remediation for cognitive impairment [70], social skills training [71], interventions to improve job skills and employment support [72], thus favoring increased patient awareness and insight, autonomy and social inclusion [20,46,65,73]. To achieve these goals, the involvement of caregivers and patients in shared decision-making on pharmacotherapy and personal needs are essential elements to improve the patient’s quality of care [46,74,75]. Furthermore, GPs’ involvement is of pivotal interest, especially in softening the physical comorbidity burdens [46,65]. Moreover, in cases of serious psychosocial functioning impairments, a rehabilitation program should be planned, even via admission into residential or semi-residential facilities [46,65]. In Italy, these interventions are provided according to community-based integrated health and social care services referring to the cost-effective model of case management, whereas additional approaches, such as assertive community treatment and intensive case management, have been found to be similarly effective [76,77].

## 2. Materials and Methods

### 2.1. Survey Construction

As we were interested in building up a survey with a multidisciplinary approach, attentive to both clinicians’ and patients’ needs, a scientific board was defined to structure the survey statements. In this phase, the scientific board included social researchers, psychologists and psychiatrists. This phase concerned a desk research design to review the existing Italian regulatory sources, guidelines and best practices on the management of mental frailties and schizophrenia [78,79,80,81,82,83,84,85]. The scientific board identified three areas of interest: *early detection and management*, *acute phase management*, *and long-term management/continuity of care*, which were considered the most significant areas in the ideal journey of a patient with schizophrenia. Then, to create the survey, the scientific board identified a list of possible statements according to the Italian regulatory sources, guidelines and best practices. 

After this, the scientific board shared the list of possible survey statements with 8 representatives of 4 patients’ and caregivers’ associations (Coplotta, Diversamente, Anpis Puglia, Club Itaca Milano) and with 3 expert peer supporter patients (aka, ESP patients). ESP patients are patients diagnosed with schizophrenia according to the ongoing classification for mental disorders and are trained at the regional level through a dedicated class to be recognized as expert peer supporters. We deemed it necessary to include ESP patients and caregivers, given their increasingly recognized relevance to patient engagement [86] and their empowerment in clinical and institutional settings. Moreover, this sharing was essential to strengthening our multidisciplinary approach, as we considered it of strategic importance to include all the stakeholders involved in an ideal patient’s journey [87]. In order to do so, a semi-structured one-on-one interview was conducted by one clinician and one social researcher with the ESP patients and patient and caregiver association representatives, with the purpose of collecting real-life evidence and relating what had emerged from the guidelines and best practices with the unmet needs still present in the management of schizophrenia.

From this interview, the scientific board was able to construct the statements that composed the present survey, focusing on specific themes that were considered of interest by clinicians, patient and caregiver association representatives and ESP patients. For *early detection and management*, we focused on services accessibility, continuity of care, multidisciplinary evaluation of patients’ needs, rehabilitation, psychoeducational and psychotherapeutic interventions, and drug treatment safety and appropriateness. For *acute phase management*, we investigated the experience of hospitalization, the prevention and decrease in commitment and forced treatment and physical restraints, and linkage to local and outpatient services. For *long-term management/continuity of care*, we focused on individual treatment plans, psychoeducational interventions, continuity in drug treatment, awareness of the patient’s physical health, recovery and social integration interventions, social and job support, and residential and semi-residential interventions. 

Thus, following a thorough validation process carried out by clinicians, ESP patients and caregivers, we finally formulated a series of statements: 17 on *early detection and management*, 16 on *acute phase management* and 27 on *long-term management/continuity of care*. Indeed, the survey finally comprised a 60-statement questionnaire built on the three main areas of interest. To answer the survey, the respondents had to express agreement or disagreement with the 60 statements on a 5-point Likert scale. Each statement was analyzed according to 2 subscales: the first subscale assessed the *importance of the statement*, from (1) “of no importance” to (5) “extremely important”; the second subscale assessed the *degree of implementation* of each statement in real-life clinical practice, from (1) “not implemented at all” to (5) “extremely implemented”. 

The survey was deployed with the CAWI (computer-assisted web interviewing) method by using a web program created and developed to manage research, surveys and customer satisfaction studies. 

### 2.2. Participants

The survey was sent to all the heads of the mental health departments (MHDs) in the Lombardy region, Italy, as an initial sample of this research. Thus, the respondents included psychiatrists only, working as heads of mental health departments (MHDs) in the Lombardy region, Italy, regardless of whether they worked in academic or non-academic settings. No patients, caregivers or other stakeholders completed the survey.

The survey was sent to 45 heads of the Lombardy MHDs, aiming to reach at least half of the responses with adequate territorial representativity: this aim was successfully achieved with 25 responses, with a 55.5% response rate. The survey was available online from 22 September 2021 to 20 January 2022, and it received the support of the Lombardy Directors of Psychiatry Steering Group and the Regional Division of the Italian Psychiatry Association. 

We supported the submission of the survey with two rounds of recalls.

### 2.3. Survey Aims

The purpose of this survey was to evaluate how the heads of MHDs could consider important topics on managing schizophrenia throughout all phases of the disorder, according to their knowledge, best practices guidelines and national regulatory sources. This was possible by analyzing the *importance of the statement* subscale. 

Then, by analyzing the *degree of implementation* subscale and the existing gaps between available knowledge/guidelines and real-life management, the survey aimed to assess how much of the available knowledge and best practices guidelines are currently applied in the Lombardy MHDs in the situation of real-life management and how MHDs could further implement the codified care paths for patients with schizophrenia.

### 2.4. Statistical Analyses

The analysis of the results comprised a general review of responses, as well as an assessment of the consensus level on the *importance of the statement* and *degree of implementation* and of the existing gaps between the guidelines and real-life clinical practice of managing schizophrenia, according to the points of view of the heads of Lombardy MHDs. 

Regarding the *importance of the statement*, a *strong consensus* was defined when rated as (4): “important” or above, whereas a low consensus was defined when rated as (3) “quite important”. Regarding the *degree of implementation*, the results were reported by combining the *degree of implementation* in 3 groups according to the mean scores for each item in the three areas of interest. A *good level of implementation* was defined for a score rated as (4) “properly implemented” or above; *moderate levels of implementation* were rated as (3) “enough implemented”; and *poor levels of implementation* were rated as (2) “slightly implemented” or below.

To quantify the consensus level on the *importance of the statement* and *degree of implementation*, we derived a mean score for each subscale for the three areas of interest and a total score. Moreover, the mode and median values were calculated. Adopting only descriptive statistical analyses, the appropriate analyses were calculated using the IBM^®^ SPSS Statistics Version 20 software. There were no a priori assumptions made. For the interpretation of the results, we primarily focused on the mean score values.

Additionally, we assessed the *gap* between the *importance of the statement* and the *degree of implementation* in order to identify which items should be further implemented with additional mental health care programs. The existence of the *gap* was defined when two conditions were satisfied: if the items of the *importance of the statement* subscale were above a score of 4, and if the items of the *degree of implementation* subscale underwent a score of 4. As the *gap* was considered when the *degree of implementation* subscale underwent a score of 4, we implicitly considered those statements showing *moderate levels of implementation,* rated as (3) “enough implemented” or below.

## 3. Results

The respondents included 25 heads of MHDs from 17 different territorial social healthcare zones of the Lombardy region: Bergamo Ovest (2), Brianza, Garda, Lecco, Lodi, Mantova, Melegnano, Martesana (2), Niguarda Milano (2), Nord Milano, Ovest Milanese (2), Papa Giovanni Bergamo, Pavia (2), Rhodense, Santi Paolo and Carlo Milano (2), Spedali Civili Brescia (3), Valcamonica and Valle Olona. 

No missing data was found, as all 25 heads of the MHDs who responded to the survey filled in all the statements.

The total results on the importance of the statement and degree of implementation are summarized in Table 1 and Table 2.

Regarding the first subscale, assessing the *importance of the statement*, a *strong consensus* emerged for several statements of the survey.

Considering the *early detection and management*, a *strong consensus* was found for all 17 statements, especially on “promotion of projects and protocols with CAMHS to promote and facilitate access to AMHS”, “to assure the continuity of care between CAMHS and AMHS”, “to create a personalized project with continuous and intensive contacts in community mental health services”, “to keep continuous and intensive contacts with family members”, “need of multidisciplinary assessment of patient’s clinical and psychosocial problems”, “to deliver a team-based multidisciplinary approach involving different healthcare professionals”, “to provide psychoeducation support”, “to provide evidence-based rehabilitation interventions”, “to provide work and study support interventions in case of moderate/severe psychosocial functioning impairment”, “to provide adequate pharmacological treatment for dosage and duration” and “to pay attention to the safety of pharmacological treatments”.

In the *acute phase management*, a *strong consensus* was found for all 16 statements, especially on “to improve accessibility to community mental health services”, “to avoid the use of physical restraint”, “to organize educational programs in order to minimize the need of physical restraint”, “to start as soon as possible an antipsychotic treatment”, “to identify the minimum effective dosage”, “to take care of safety of pharmacological treatment through an early monitoring of side effects”, “maintenance of pharmacological treatment for at least two years at discharge”, “to ensure a continuity of care with community MHS”, “ to ensure intensive contacts with community MHS after discharge” and “to review the ongoing treatment plans, when an hospitalization occurs, through a collaboration between inpatient and outpatient healthcare services”.

In the *long-term management/continuity of care*, a *strong consensus* was obtained for 26 out 27 statements, especially on “to provide continuous and multidisciplinary-based treatment to promote full psychosocial recovery”, “to define an individual treatment plan and to identify a case manager”, “to take care of family members”, “to provide psychoeducational and psychotherapeutic treatments for patients”, to provide psychoeducational treatments for family members”, “to carefully assess and treat substance abuse disorders conjointly with dedicated addiction services”, “to evaluate physical health conjointly with GPs”, “to monitor patient’s lifestyle in collaboration with GPs”, “to offer clozapine in case of treatment-resistance”, “to offer LAI antipsychotic treatment in case of frequent relapses and poor adherence”, “to maintain regular contact with patients that interrupted drug treatment”, “to re-contact patients who interrupted the contact with outpatients mental health service”, “to promote peer support groups oriented to recovery and social inclusion”, “to monitor adverse outcomes of the patients being cared for (death, suicide)”, “to provide psychosocial interventions and work placement support”, “to provide evidenced-based rehabilitation and resocialization interventions either in community and day-care facilities” and “to provide rehabilitation programs in residential facilities in case of serious psychosocial functioning impairment aiming for the patient’s return at home”. Only one statement (e.g., “to offer psychotherapeutic treatments for patients’ relatives and family members”) was ranked as (3) “quite important”. 

The second subscale of the survey assessed the *degree of implementation*.

*Good levels of implementation* were found on 8 out of 17 statements (47% of the sample) for the *early detection and management* area, on 14 out of 16 statements for the *acute phase management* area (87.50% of the sample), and on 15 out of 27 statements for the *long-term management/continuity of care* area (55.6% of the sample).

*Poor levels of implementation* were found on 1 out of 17 statements of *early detection and management* (6% of the sample), while no statements were rated as poorly implemented neither for *acute phase management* nor *long-term management/continuity of care*. 

Figure 1 shows the number of items (and the percentage of the total items for each of the three thematic areas) divided according to the degree of implementation.

Focusing on each item of the *early detection and management*, *good levels of implementation* were especially found on “to provide adequate pharmacological treatment for dosage and duration” and “to pay attention to the safety of pharmacological treatments”.

Focusing on each item of the *acute phase management*, *good levels of implementation* were especially found on “to start as soon as possible an antipsychotic treatment”, “to take care of safety of pharmacological treatment through an early monitoring of side effects”, “maintenance of pharmacological treatment for at least two years at discharge” and “to ensure a continuity of care with community MHS”.

Focusing on each item of the *long-term management/continuity of care*, *good levels of implementation* were especially found on “to define an individual treatment plan and to identify a case manager”, “to offer clozapine in case of treatment-resistance” and “to offer LAI antipsychotic treatment in case of frequent relapses and poor adherence”.

On the other hand, *poor levels of implementation* were found in the *early detection and management* phase, on “promoting projects and protocols with GPs aimed at prevention”.

Then, we considered the *gap* between the *importance of the statement* and the *degree of implementation* by considering those statements showing *moderate levels of implementation* or below.

Considering the *early detection and management*, *moderate levels of implementation* were found on 8 out of 17 statements (47% of the sample), especially on the “promotion of projects and protocols with CAMHS to promote and facilitate access to AMHS”, “to assure the continuity of care between CAMHS and AMHS”, “to use internationally validated assessment tools”, “to assess family burdens and their needs”, “to deliver multidisciplinary support for family members”, “to provide home interventions”, “to provide psychotherapy interventions” and “to provide work and study support interventions in case of moderate/severe psychosocial functioning impairment”. Figure 2 summarizes the mean scores of the *importance of the statement*, *degree of implementation*, and the *gap* between these subscales for the items related to *early detection and management*.

Considering the *acute phase management*, *moderate levels of implementation* were found on 2 out of 16 statements (12.50% of the sample), especially on the “need of acute inward admission when an acute decompensation occurs” and on the “need to limit pharmacological restraint”. Figure 3 summarizes the mean scores of the *importance of the statement*, *degree of implementation*, and the *gap* between these subscales for the items related to *acute phase management*.

Concerning the *long-term management/continuity of care*, *moderate levels of implementation* were found on 12 out of 27 statements (44.4% of the sample), especially on “to take care of family members”, “to provide psychoeducational and psychotherapeutic treatments for patients and for family members”, “to evaluate physical health conjointly with GPs”, “to monitor patient’s lifestyle in collaboration with GPs”, “to promote peer support groups oriented to recovery and social inclusion”, “to promote the integration of expert in peer support”, “to promote the role of the expert in peer support in improving efficacy of treatments” and “to promote rehabilitation programs in semi-residential facilities for patients with a good level of autonomy”. Figure 4 summarizes the mean scores of the *importance of the statement*, *degree of implementation*, and the *gap* between these subscales for the items related to the *long-term management/continuity of care*.

Figure 5 summarizes the overall mean scores of the *importance of the statement*, *degree of implementation*, and the *gap* between these subscales for the three areas of interest.

For *early diagnosis and management*, while a *strong consensus* was found (mean score = 4.63), the level of implementation was found to be slightly below the cut-off (mean score = 3.97), being rated as moderate-to-good. Therefore, we can assume that only a minor gap emerged in this area of interest.

For the *acute phase management*, a *strong consensus* (mean score = 4.69) and a *good level of implementation* (mean score = 4.30) were found. 

For *long-term management/continuity of care*, while a strong consensus was found (mean score = 4.62), the level of implementation was found to be slightly above the cut-off (mean score = 4.02).

Overall, the survey found a *strong consensus* (mean score = 4.63) and a *good level of implementation* (mean score = 4.07) for the analyzed statements.

## 4. Discussion

Schizophrenia represents a leading cause of disability worldwide. Indeed, people living with schizophrenia present with poor quality of life, insufficient psychosocial functioning, increased unemployment levels, social isolation and a reduced life expectancy due to excess mortality and morbidity. Thus, early recognition and appropriate management are needed to reduce the risk of chronicity and comorbidity, especially in the case of dual diagnosis. Thus, the personalization and integration of pharmacological and psychosocial interventions, as well as the accurate identification and management of psychiatric and somatic comorbidities, can significantly improve the mental and physical health of patients living with schizophrenia, thus promoting recovery [73]. In this scenario, both at a national and local level, MHSs should identify the strengths and weaknesses of their interventions in order to promote constant updating of the organization and implementation of the MHS. Indeed, a previous Delphi study found several weaknesses (i.e., lack of time, human resources and training) as the main barriers and challenges to the translation of knowledge into clinical practice [20]. Moreover, as healthcare utilization (HCU) databases can reveal the strengths and weaknesses of the national care system by representing a useful tool in the routine assessment of mental healthcare quality [79], a recent multi-regional Italian investigation based on the HCU databases found that to reduce regional variability, Italian MHSs should improve the accessibility to psychosocial interventions and the quality of care for newly taken-in-care patients, focusing on somatic health and mortality [88]. 

Thus, with the Patient Journey Project, we built a survey considering three macro areas (*early detection*, *acute phase management* and *long-term management/continuity of care*), with the contribution of all the stakeholders involved in the planning of the ideal clinical path of care, including clinicians, expert patients, caregivers and family members, in order to describe the current evidence and to discuss the unmet needs of the care management of schizophrenia according to Lombardy MHD heads’ point of views. Thus, covering these macro areas, we will discuss the consensus on the importance of many areas of intervention, analyze the areas with good or low levels of implementation, and also discuss the gaps between the importance of the statements and their implementation in Lombardy’s psychiatric services.

Regarding *early detection and management*, the survey highlighted a *strong consensus* concerning the importance of promoting shared protocols to facilitate access from CAMHS to AMHS, assuring continuity of care between child and adult services, and also keeping continuous and intensive contact with family members. However, the survey found that these protocols are only implemented enough among the Lombardy MHDs: these findings suggest further developing structured plans favoring the transition and continuity of care from CAMHS to AMHS. Indeed, while national and international guidelines strongly recommend this approach, in Lombardy, only a few examples are available in clinical practice. Among them, the mental health department of the territorial social healthcare zones of Melegnano and Martesana conducted a four-year follow-up study to verify the continuity of care effectiveness for adolescent patients aged between 16 and 19 who were transitioning from CAMHS to AMHS. Of the 93 users, most cases (54.8%) needed to be managed by a multidisciplinary team in order to have an effective transition. Moreover, when a dedicated team was arranged, the success of the transition was achieved in 86% of the cases; therefore, 88.6% of the patients were still followed one year after their transition from CAMHS to AMHS [89]. These results highlighted that an effective transition from one service to another is possible if this process is adequately structured and planned and possibly delivered by a specialized and dedicated team. 

Moreover, the creation of an individualized project—one that involves continuous and intensive contact between community mental health services and the patient and their family—represents another core need that is properly implemented within the Lombardy MHDs. In this light, the need for a multidisciplinary assessment of a patient’s clinical and psychosocial problems by a team-based multidisciplinary approach is another crucial aspect to consider during the early phases of the patient’s journey, especially with the aim of providing work and study support interventions in case of severe psychosocial functioning impairment. Nevertheless, while the survey showed a *strong consensus* for these statements and *good levels of implementation*, in Lombardy, only a few pilot projects have been carried out in public MHSs, including the Ambulatorio Spazio Giovani [90] and Centro Giovani “Ponti” [91], with the aim of protecting the mental health of young adults, thus suggesting that further efforts are urgently required to bridge this gap.

In addition, while a good consensus was found regarding the importance of employing internationally validated assessment tools during the early detection phase, the implementation level among MHDs was considerably low. Thus, a further suggestion is that practitioners should be routinely trained on the application of these tools, including the *Comprehensive Assessment of At-Risk Mental States (CAARMS)* that assesses both the psychopathology and identification of individuals at high risk for psychosis [92], the *Structured Interview for Prodromal Symptoms/Structured Interview for Psychosis-Risk Syndromes (SIPS)* [93], and the *Kiddie Schedule for Affective Disorders and Schizophrenia (K-SADS)*, which is more generally aimed at an early diagnosis of both psychotic and anxious/affective disorders in youths [94]. The standardization of assessment tools represents an unresolved matter in psychiatry, as the diagnostic evaluation could be extremely heterogeneous, complicating the evaluation of the psychopathological needs of subjects in a prodromal phase. To this extent, the application of shared protocols could reduce this heterogeneity, and through the incorporation of all the information from family members or the real-life context, it might represent a timely intervention that avoids further diagnostic and therapeutic delays and clinical deterioration.

In line with national and international guidelines and recommendations [24,46,47,48,73] relating to therapeutic interventions, a *strong consensus* and *good levels of implementation* were both observed in the present survey regarding the provision of adequate pharmacological treatment in terms of the congruent dosage and duration, with a special focus on safety and tolerability profiles. Conversely, several non-pharmacological interventions (i.e., home-based interventions, psychotherapy, psychoeducation, rehabilitation and work/study support interventions) were found to be of great importance in the survey. However, important gaps emerged regarding their implementation in real-world clinical practice among the Lombardy MHDs, as these interventions were at a moderate level of implementation.

Moreover, while satisfactory consensus emerged regarding GPs’ involvement during the *early detection and management* phase, severe discordance was found regarding its implementation in clinical practice. Nevertheless, some reports highlighted that GPs considered themselves as the first line of contact with cases experiencing acute symptoms of psychosis, providing support and psychoeducation to the patient’s family, thus not limiting their action to somatic comorbidity management [95,96]. Although primary care for psychotic patients depends on the GP’s personal resources, as the quality of collaboration between GPs and MHSs represents another key factor in moderating the GP’s engagement in the patient’s journey [97], further efforts are needed to develop shared plans and connections between GPs and mental health providers to improve the quality of care among adolescents with a constraining psychotic disorder as well as during long-term management [98]. 

Finally, the survey found that topics concerning the family’s involvement (i.e., the assessment of family burden and the multidisciplinary support provided to family members) are only at *moderate levels of implementation* among Lombardy MHDs. These points emphasize the need to further realize plans to reduce the family burdens of those young patients who are in the early stages of the disease. A suggestion could be to promote further connections with GPs, AMHSs and family members through public awareness campaigns, social media and public events.

Summing up, as we found moderate-to-good levels of implementation in *early diagnosis and management*, we could assume that further efforts are needed to improve the patient’s journey during the early phases: in particular, the Lombardy MHDs should focus on the transition from CAMHS to AMHS and on the involvement of GPs and family members to increase the efficacy of preventive actions.

On the other hand, the survey showed an overall *strong consensus* for all the statements dedicated to *acute phase management*. Notably, a great consensus was reached regarding the provision of adequate antipsychotic treatments in terms of their immediate availability during a crisis while also taking into consideration the safety and tolerability profiles of the prescribed drugs. As international recommendations suggested [55,56,73], prior to choosing an antipsychotic, a clear recommendation is to evaluate the clinical presentation and the somatic comorbidities to further improve clinical efficacy, tolerability and long-term adherence.

Furthermore, attention should be paid when an acute admission occurs. Indeed, hospitalization should be conceived not only as a strategy to manage the psychopathological crisis through a timely antipsychotic treatment at the minimum effective dose but also as a background in which diagnostic assessment of somatic problems should be required and guaranteed through a feasible collaboration with other medical disciplines [99], while also considering the safety of the pharmacological treatments through the early monitoring of side effects. 

In addition, participants in the survey found a *strong consensus* and quite satisfactory implementation levels regarding the role of acute admission, ensuring a rapid continuity of care with outpatient services. Thus, acute hospitalization could represent a crucial framework during which a re-evaluation of treatment plans should be recommended, especially in those cases of poor clinical and psychosocial outcomes and refractory symptoms, anticipating the unmet needs of the patients, caregivers and outpatient community services [44,46,98]. Similarly, the survey found a satisfactory consensus on the theme appropriateness of hospitalization to manage the acute phase of a psychotic episode, emphasizing how difficult it is for the MHSs to adequately implement outpatient care systems to prevent acute hospitalization. Indeed, this practice was found to be only *enough implemented* among Lombardy MHDs, indicating that MHSs should develop flexible strategies aimed at preventing emergencies through the implementation of outpatient and community-based services, ensuring continuity of care. Accordingly, these plans should consider proper community-based interventions for medium-to-long-term periods, involving all the figures already active in the treatment process, including GPs, local mental health providers, the patients and their relatives [44,98]. With a view to reconsidering the hospital-community relationship, a suggestion could be to avoid inappropriate admissions, especially in those cases of social problems or when odd behaviors that are not primarily attributable to psychopathological frameworks are present [99]. Moreover, to ensure continuity of care, a great consensus and *good implementation levels* were found regarding the need to provide quick and intensive contact with community mental health services following hospital discharge and on the maintenance of pharmacological treatment in the follow-up.

Moreover, a *strong consensus* was reached regarding the suggestion to avoid coercive treatments during hospitalization, including involuntary admissions and physical/pharmacological restraints, with a good level of implementation to obtain this. Notably, the Lombardy MHD heads have expressly suggested the adoption of specific protocols aimed at limiting, preventing and managing the possible applications of physical restraint measures [98]. Therefore, great importance is given to educational programs, where meta-analytic evidence suggests that when educational programs are provided for mental health workers, the use of physical restraint is significantly reduced [99]. Moreover, the duration and frequency of education produced a stronger effect in reducing the use of physical restraints, thus suggesting that a continuous education program, especially involving nurses, should be provided to prevent physical and pharmacological restraint [98,99,100]. Nevertheless, while participants in the survey reported a good consensus in limiting the use of pharmacological restraints, only *moderate levels of implementation* were found: these results are not surprising since several factors seem to be associated with increased levels of restraints, including sociodemographic factors (i.e., male gender, younger age, foreign ethnicity) [101,102], the type of hospitalization (e.g., compulsory admissions) [103,104], type of diagnosis (e.g., substance abuse, psychotic or affective disorders) [103,104] and environmental factors (e.g., the presence of male staff, night time, a seasonal trend with frequent pharmacological coercion during spring and mechanical coercion during summer) [101,103,104,105]. 

Finally, the survey highlighted a *strong consensus* for all the statements regarding *long-term management/continuity of care*, except for the provision of psychotherapy treatment for family members. In particular, a high consensus was achieved regarding the need to provide continuous and multidisciplinary-based treatment to promote full psychosocial recovery, define an individual treatment plan, identify a case manager, provide psychoeducational and psychotherapy treatments, and carefully assess concomitant substance abuse disorders conjointly with addiction services, as well as offer LAI treatments to reach great adherence, offer clozapine in case of treatment resistance, and provide rehabilitation programs in residential facilities for serious psychosocial functioning impairment. These results are expected, as the Lombardy operating model for people living with schizophrenia gave particular attention to long-term management, especially in those cases of high clinical complexity, and always with the aim of ensuring continuity of care through various outpatient services and the activation of familiar, social and environmental resources. This is possible with the creation of a multi-disciplinary community-based intervention through the definition of the so-called individual treatment plan and the application of the case management approach [44,46,98]: starting with a systematic analysis of the patient’s complex needs, both at a psychopathological and psychosocial level, through a multi-professional team-based evaluation. This operating model has made it possible to improve the quality and management of psychiatric assistance, thus reducing interventions in emergency situations and allowing for the adequate planning of future treatment paths to achieve recovery [99].

Nevertheless, the survey also found several gaps between the importance of statements and the degree of implementation in clinical practice for *long-term management/continuity of care*. Indeed, while the level of implementation was found to be slightly above the cut-off, almost 45% of the statements (that is, 12/27) were rated as *enough implemented*. Specifically, *moderate levels of implementation* were found for psychoeducational and psychotherapeutic treatments for patients and family members, collaborations with GPs who monitor the patient’s physical health and lifestyle, the provision of peer support groups together with the collaboration of experts and multi-professional teams to improve treatment efficacy, achieve recovery and social inclusion, and the provision of rehabilitation plans in semi-residential facilities, including social housing. These problems have already been emphasized in the *Definizione dei percorsi di cura da attivare nei Dipartimenti di Salute Mentale per i Disturbi schizofrenici, i Disturbi dell’umore e i Disturbi gravi di personalità* agenda [65], where it was recommended to further manage these issues. After almost a decade, the Lombardy MHDs must clearly provide further indications for developing and implementing adequate interventions to respond to these structural problems. While evidenced-based rehabilitation programs in residential facilities were found to be adequately improved in the survey, one suggestion is to rethink the role of semi-residential facilities, the so-called day centers. In fact, semi-residential structures are designated to solve therapeutic-rehabilitative functions, including pharmacological intervention to prevent and contain hospitalization [65]. Moreover, day centers are placed in the community context. A multi-professional team can help the patient experiment and learn skills or strategies to implement psychosocial functioning, including self-care, daily life activities and interpersonal relationships, while also giving employment advice [44,46,106,107]. In this context, the day centers could be a feasible solution in supporting an effective continuity of care in the community setting, counterbalancing the phenomenon of interminable psychiatric residency by favoring rehabilitation, socialization and social reintegration, thanks to a strong link with primary and secondary community networks [107], thus also reducing the high levels of familiar expressed emotions [107]. Moreover, when engaged in day center activities, ESP patients could be of pivotal importance in promoting peer support to improve recovery, social inclusion and treatment efficacy.

In summary, the present survey offers an updated evaluation of the intervention areas of priority importance for MHD heads and also covers the current limitations of the Lombardy MHSs that affect various elements of the journey of patients with schizophrenia. In particular, the moment of transition from CAMHs to AMHs during the early phase and the management of chronicity (with particular mention to the psychological support for family members, the provision of semi-residential interventions, the monitoring of patients’ lifestyles and physical health in collaboration with their GPs and the integration of experts in peer support in a multi-professional team) represent the current unmet needs to which the Lombardy MHSs must respond to further improve the patient’s journey of living with schizophrenia.

Thus, several issues could represent new areas for future interventions. Firstly, as aerobic exercise seems to improve cognitive functioning among people with schizophrenia [108,109] and physical activity and a person’s well-being represent key moderators of primary prevention and clinical treatment [110], MHSs should develop systematic plans to improve lifestyle and physical health, particularly focusing on tobacco smoking cessation, healthy diets and the promotion of sleep-focused interventions [111]. This can be achieved by encouraging further integration of the MHSs into primary care settings and providing supervised exercise to people with schizophrenia [110].

Secondly, current evidence suggests that only an integrated approach involving pharmacotherapy and psychosocial interventions could improve outcomes in schizophrenia to achieve recovery. Thus, the efficacy of several psychosocial interventions, including approaches such as cognitive rehabilitation [70], CBT [111], social skill training [71], illness self-management training and supported employment [112], is supported by substantial evidence in many outcomes measures, and therefore, a further suggestion for the Lombardy MHDs is to systematically integrate these evidence-based approaches into the daily management of people living with schizophrenia [113]. Moreover, as family intervention and psychoeducation seem to reduce the number of relapse events and hospitalizations [114,115,116,117], MHDs should further promote and integrate family care approaches in the long-term management of patients with schizophrenia through the participation of relatives in such psychoeducational programs, thus helping to build a proper family emotional environment that will benefit the caregiver–patient relationship and improve patient functioning [117]. 

Thirdly, peer support is an essential component of the patient’s journey for achieving personal recovery that MHDs should improve on in daily clinical practice: based on the rationale that people who have experienced mental illness [118] are qualified to provide support and hope to others dealing with similar challenges, peer support programs with ESP patients focus on helping others to become active participants in their recovery process, breaking out of the passive role of the patient [113,119]. Moreover, by becoming involved in the recovery process of others, peers who provide support represent one of the best examples of community integration, increasing personal autonomy and self-worth [115]. Lastly, an important factor for people living with schizophrenia is a personal participation in the definition of their care pathway [73]. Indeed, patients require autonomy and participation in treatment decisions, including active participation in the so-called shared decision-making (SDM) process in treatment choice [74]. Current evidence has provided good preliminary data on SDM as a method to improve mental health services, including guideline-concordant care, attendance and retention in treatment, and satisfaction with health care, thus representing a promising strategy to improve collaboration between clinicians and patients in achieving recovery [120]. Notably, there is evidence that SDM is feasible and time-comparable to the usual care in psychiatric and primary care settings when considering both acute [121] and long-term drug treatment choices [75]. Clinicians should evaluate the patient’s preferences, expectations and concerns towards the development of a personalized treatment strategy in order to reduce the patient’s perception of being forced to receive a drug [75], thus achieving greater involvement in the treatment process and increasing knowledge about the illness [74]. 

Lastly, another suggestion could be to improve the family’s support and the patient’s engagement with cultural and ethical practices. Indeed, it was suggested that the perceived role of family bonds and religiosity–spirituality could positively influence clinical outcomes, such as treatment maintenance-adherence and mortality [122,123]: this could be partly linked to improved health practices, increased social contacts and reduced stigma levels [122]. Thus, an appropriate analysis of these psychosocial factors, including cultural and ethnic milieu, could be helpful to further improve the patient’s journey.

In this project, some limitations are to be addressed. First, the results cover only the Lombardy MHDs; thus, they are not representative of the entire national territory and may not be generalizable to other countries. Second, the survey involved only specialists in adult psychiatry, not child neuropsychiatrists, GPs and other mental health workers. Therefore, our study does not cover the views of all the stakeholders involved in this complex field. 

## 5. Conclusions

Despite these limitations, the present survey, co-designed by clinicians, expert patients and caregivers, offers an updated evaluation of the areas of intervention of priority importance for the patient’s journey with schizophrenia in MHSs, also covering several current critical issues. Particularly, the moment of transition from child to adult services in the early phase of the disorder and the management of chronicity represent the unmet needs to which a response is needed to improve the patient’s journey when living with schizophrenia.

## Figures and Tables

**Figure 1 brainsci-13-00822-f001:**
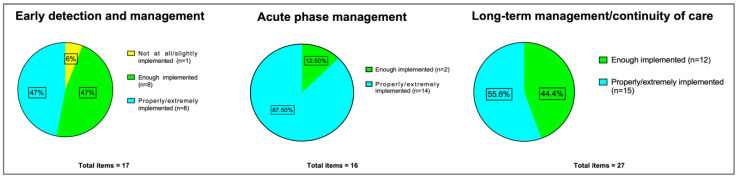
Degree of implementation. The figure shows the number of items (and the percentage of the total items for each of the three thematic areas) divided according to the level of implementation. The results of the *degree of implementation* subscale are subdivided into 3 groups according to the mean scores for each item in the three areas of interest. *Good level of implementation* was defined for a score rated as (4) “properly implemented” or above; *moderate levels of implementation* was rated as (3) “enough implemented”; *poor levels of implementation* was rated as (2) “slightly implemented” or below.

**Figure 2 brainsci-13-00822-f002:**
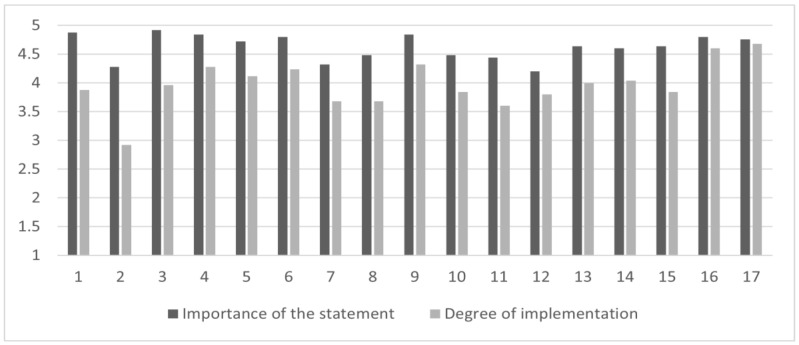
The *gap* between the importance of the statement and the degree of implementation in the *early detection and management* area. Figure legend: the abscissa axis represents items (0–17) of the *early detection and management*; the ordinate axis represents the 5-point Likert scale anchor points: deep gray for *importance of the statement* (from (1) “of no importance” to (5) “extremely important”) and light gray for *degree of implementation* (from (1) “not implemented at all” to (5) “extremely implemented”).

**Figure 3 brainsci-13-00822-f003:**
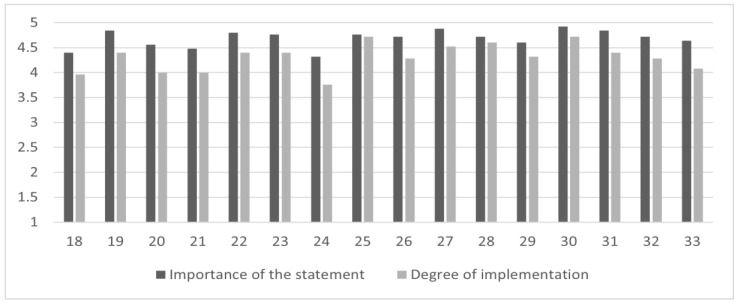
The *gap* between the importance of the statement and the degree of implementation in the *acute phase management* area. Figure legend: the abscissa axis represents items (18–33) of the *acute phase management*; the ordinate axis represents the 5-point Likert scale anchor points: deep gray for *importance of the statement* (from (1) “of no importance” to (5) “extremely important”) and light gray for *degree of implementation* (from (1) “not implemented at all” to (5) “extremely implemented”).

**Figure 4 brainsci-13-00822-f004:**
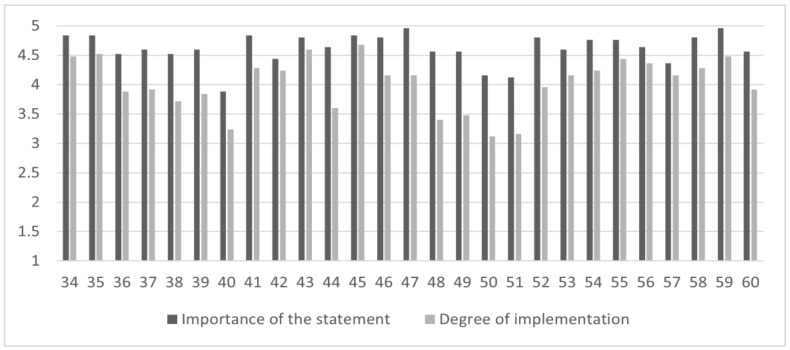
The *gap* between the importance of the statement and the degree of implementation in the *long-term management/continuity of care* area. Figure legend: the abscissa axis represents items (34–60) of the *long-term management/continuity of care* area; the ordinate axis represents the 5-point Likert scale anchor points: deep gray for *importance of the statement* (from (1) “of no importance” to (5) “extremely important”) and light gray for *degree of implementation* (from (1) “not implemented at all” to (5) “extremely implemented”).

**Figure 5 brainsci-13-00822-f005:**
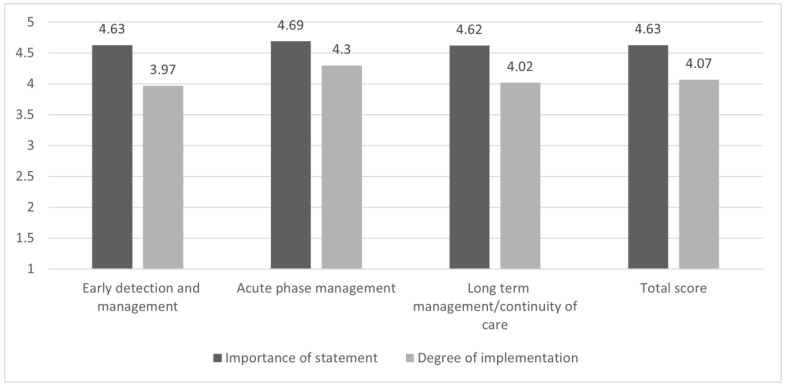
The *gap* between the importance of the statement and the degree of implementation (mean scores). The figure summarizes the overall mean scores of the *importance of the statement*, *degree of implementation*, and the *gap* between these subscales for the three areas of interest and the total score. Deep gray for *importance of the statement* (from (1) “of no importance” to (5) “extremely important”) and light gray for *degree of implementation* (from (1) “not implemented at all” to (5) “extremely implemented”).

**Table 1 brainsci-13-00822-t001:** Importance of statement and degree of implementation (mean scores).

	Importance of the Statement	Degree of Implementation
**Early Detection and Management**		
(1) Projects and protocols with CAMHS to promote access to AMHS	4.88	3.88
(2) Projects and protocols with GPs aimed at prevention	4.28	2.92
(3) Continuity of care between CAMHS and AMHS	4.92	3.96
(4) Personalized project with continuous and intensive contacts in community mental health services	4.84	4.28
(5) Continuous and intensive contacts with family members	4.72	4.12
(6) Multidisciplinary assessment of patient’s clinical and psychosocial problems	4.8	4.24
(7) Using internationally validated and widespread assessment tools	4.32	3.68
(8) Assessment of family burden and their needs	4.48	3.68
(9) Team-based multidisciplinary approach involving different healthcare professionals	4.84	4.32
(10) Multidisciplinary support to family members	4.48	3.84
(11) Home interventions	4.44	3.6
(12) Psychotherapy	4.2	3.8
(13) Psychoeducation	4.64	4
(14) Rehabilitation	4.6	4.04
(15) Work and study support interventions	4.64	3.84
(16) Adequate pharmacological treatment for dosage and duration	4.8	4.6
(17) Safety of pharmacological treatment	4.76	4.68
*Total score*	4.63	3.97

**Acute Phase Management**		
(18) Not necessary in acute inward admission	4.4	3.96
(19) Improve accessibility to community mental health services	4.84	4.4
(20) Paying attention to the emotive impact of hospitalization	4.56	4
(21) Reduce involuntary admission	4.48	4
(22) Avoid the use of physical restraint	4.8	4.4
(23) Educational programs in order to minimize the need for physical restraint	4.76	4.4
(24) Limit pharmacological restraint	4.32	3.76
(25) Antipsychotic treatment as soon as possible	4.76	4.72
(26) Minimum effective dosage	4.72	4.28
(27) Safety of pharmacological treatment	4.88	4.52
(28) Maintenance of pharmacological treatment for at least two years	4.72	4.6
(29) Limit duration of hospitalization	4.6	4.32
(30) Ensure rapid continuity of care with the community mental health services	4.92	4.72
(31) Intensive contact with community mental health service after discharge	4.84	4.4
(32) Review of the treatment program during hospitalization among inpatient and outpatient healthcare professionals	4.72	4.28
(33) Review of the treatment program between hospitalized patients and caregivers of the community mental health service	4.64	4.08
*Total score*	4.69	4.30

**Long-Term Management/Continuity of Care**		
(34) Continuous and multidisciplinary-based treatment	4.84	4.48
(35) Define an individual treatment plan identifying a case manager	4.84	4.52
(36) Take care of the family members	4.52	3.88
(37) Psychoeducational treatment for patients	4.6	3.92
(38) Psychoeducational treatment for family members	4.52	3.72
(39) Psychotherapeutic treatment for patients	4.6	3.84
(40) Psychotherapeutic treatment for family members	3.88	3.24
(41) Carefully managing substance abuse disorders with the help of addiction services	4.84	4.28
(42) Monotherapy antipsychotic treatment	4.44	4.24
(43) Clozapine in case of treatment resistance	4.8	4.6
(44) Evaluate physical health in collaboration with GPs	4.64	3.6
(45) Long-acting injectable antipsychotic treatment for patients with frequent relapses and poor adherence	4.84	4.68
(46) Regular contact with patients who stop drug treatment	4.8	4.16
(47) Re-contact patients who interrupted contact with the community mental health service	4.96	4.16
(48) Monitoring of patients’ lifestyles in collaboration with GPs	4.56	3.4
(49) Peer support groups oriented to recovery and social inclusion	4.56	3.48
(50) Integration of the expert in peer support in a multi-professional team	4.16	3.12
(51) Role of the expert in peer support in improving the efficacy of treatments	4.12	3.16
(52) Monitoring of adverse outcomes (death, suicide)	4.8	3.96
(53) Assessment of patients’ job skills	4.6	4.16
(54) Psychosocial interventions and work placement actions	4.76	4.24
(55) Evidenced-based rehabilitation interventions, either in the community or day-care facilities	4.76	4.44
(56) Resocialization interventions, either in the community or day center facilities	4.64	4.36
(57) Residential facilities in case of serious psychosocial functioning impairment	4.36	4.16
(58) Rehabilitation programs in residential facilities in case of serious psychosocial functioning impairment	4.8	4.28
(59) Rehabilitation programs in residential facilities aimed at patient’s return to home	4.96	4.48
(60) Rehabilitation programs in semi-residential facilities for patients with a good level of autonomy	4.56	3.92
*Total score*	4.62	4.02

*Mean score*	4.63	4.07

AMHS, Adult Mental Health Services; CAMHS, Child and Adolescent Mental Health Services; GPs, general practitioners.

**Table 2 brainsci-13-00822-t002:** Importance of statement and degree of implementation (mean scores, medians, mode and standard deviations).

Importance of the Statement				Degree of Implementation		
	Early Detection and Management	Acute Phase Management	Long-Term Management Continuity of Care	Early Detection and Management	Acute Phase Management	Long-Term Management Continuity of Care
**Mean**	4.63	4.69	4.62	3.97	4.30	4.02
**SD**	0.605	0.558	0.622	0.967	0.817	0.971
**Median**	5	5	5	4	4	4
**Mode**	5	5	5	4	5	5

SD, standard deviations.

## Data Availability

The data presented in this study are available on request from the corresponding author.

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
