# Peer review of "The Patient Journey of Schizophrenia in Mental Health Services: Results from a Co-Designed Survey by Clinicians, Expert Patients and Caregivers"

_brainsci, 2023, doi:10.3390/brainsci13050822_

Round 1

Reviewer 1 Report

The article explored experiences in clinical practice in the context of the Patient Yourney project concerning patients suffering from schizophrenia. The topic is original and relevant because there are no studies dealing with this specific sample in Italy. The title corresponds to the content, and the article does not need additions or shortening. The authors use understandable language, and the summary is appropriate to the content. The content is very well organised, and it is evident that the authors have a good knowledge of the latest research when it comes to the mentioned disease and ways of dealing with it. The authors use terms, methods, and terminology precisely. The objectives, methods, results, and conclusions are very well connected, and the style is clear and concise.

Author Response

Reviewer #1

We thank reviewer #1 for taking the time to evaluate the manuscript and appreciating it so much.

We hope that paper like this could help to further improve the organization of mental health services and the management of schizophrenia both at regional and national level.

Reviewer 2 Report

This is a detailed article describing an interesting topic. The author(s) can additionally work on some points such as:

In the introduction, please add a separate section about the investigated topic and data from Italy, such as Clerici, M., De Bartolomeis, A., De Filippis, S., Ducci, G., Maremmani, I., Martinotti, G., & Schifano, F. (2018). Patterns of management of patients with dual disorder (psychosis) in Italy: a survey of psychiatrists and other physicians focusing on clinical practice. Frontiers in Psychiatry9, 575.

Galderisi, S., Riva, M. A., Girardi, P., Amore, M., Carpiniello, B., Aguglia, E., ... & Vita, A. (2020). Schizophrenia and “unmet needs”: From diagnosis to care in Italy. European Psychiatry63(1), e26.

100% response rate is peculiar. Please explain.

In the discussion, please take into consideration a relevant comment on the issue of severe mental illness in different cultural settings and for different ethnicities even in the same country: Giannouli, V. (2017). Ethnicity, mortality, and severe mental illness. The Lancet Psychiatry4(7), 517. You do not acknolwedge this point in your text, but it needs to be discussed.

In the methods section please state clearly the characteristics of the participants.

The results need to be presented in a way that is clear to the reader. In this current form, they are detailed, but difficult to follow. Please rephrase.

Author Response

Reviewer #2

  • In the introduction, please add a separate section about the investigated topic and data from Italy, such as Clerici, M., De Bartolomeis, A., De Filippis, S., Ducci, G., Maremmani, I., Martinotti, G., & Schifano, F. (2018). Patterns of management of patients with dual disorder (psychosis) in Italy: a survey of psychiatrists and other physicians focusing on clinical practice. Frontiers in Psychiatry9, 575. Galderisi, S., Riva, M. A., Girardi, P., Amore, M., Carpiniello, B., Aguglia, E., ... & Vita, A. (2020). Schizophrenia and “unmet needs”: From diagnosis to care in Italy. European Psychiatry63(1), e26.

We thank reviewer #2 for these important suggestions to improve the quality of the manuscript.

We added in the introduction a section on the theme of the dual diagnosis according to the work by Clerici et al., 2018, as follows:

“Moreover, great burdens also derived from the so-called Dual Diagnosis condition, that this when patients affected by severe psychiatric disorders suffer from concomitant substance use disorders (SUD). Indeed, data from an Italian study showed that the management of comorbid SUD in patients with schizophrenia is increasingly complex, highlighting an urgent need to optimize the management of this difficult-to-treat condition by considering several factors (i.e., treatment efficacy, tolerability, metabolic effect sides) and according to a multidisciplinary approach throughout all the phases of both disorders [19]”.

This theme was also considered in the Discussion, as in the following sentence:

“Thus, early recognition and appropriate management is needed to reduce the risk of chronicity and comorbidity, especially in the case of dual diagnosis”.

As for the paper by Galderisi et al. 2020, we believe that this work is of pivotal interest for our purposes. Indeed, it has already cited throughout all the section of the paper, especially at the beginning of the Discussion. However, we moved these argumentations in the Introduction, as requested by reviewer #2, as follows:

“Indeed, a previous Delphi study explored the consensus of Italian experts, psychiatrists, and trainees in psychiatry, and showed high consensus on several components of schizophrenia care, including early recognition, personalization, and integration of care, assessment standardization, management of somatic and psychiatric comorbidities [20]”.

Then, we rephrased a sentence in the Discussion in the following way:

“Indeed, a previous Delphi study found several weaknesses (i.e., lack of time, human resources, and training) as the main barriers and challenges to the translation of knowledge into clinical practice [20]”.

  • 100% response rate is peculiar. Please explain.

This a mere mistake and we apologize for this inaccuracy. We rephrased the sentence as follows:

“The survey was sent to 45 Heads of Lombardy MHDs, aiming to reach at least half of the responses with adequate territorial representativity: this aim that was successfully achieved with 25 responses, with a 55.5% response rate”.

  • In the discussion, please take into consideration a relevant comment on the issue of severe mental illness in different cultural settings and for different ethnicities even in the same country: Giannouli, V. (2017). Ethnicity, mortality, and severe mental illness. The Lancet Psychiatry4(7), 517. You do not acknolwedge this point in your text, but it needs to be discussed.

We thank reviewer #2 for this important suggestion that was not previously addressed in the paper. To overcome this limit, we added the following paragraph in the Discussion:

“Lastly, another suggestion could be to improve family support and patient’s engagement with cultural and ethnical practices. Indeed, it was suggested that the perceived role of family bonds and of religiosity - spirituality could positively influence clinical outcomes such as treatment maintenance-adherence and mortality [126-127]: this could be partly linked to improved health practices, increased social contacts and reduced stigma levels [126]. Thus, an appropriate analysis of these psychosocial factors, including cultural and ethnical milieu, could be helpful to further improve the patient’s journey”.

  • In the methods section please state clearly the characteristics of the participants.

We thank the reviewer #2 for this insightful comment to increase the clarity of the manuscript. We created a separated section in the Methods (2.2 participants) with the aim to address the reviewer’s suggestion. Then, we clearly stated the characteristics of the participants, as follows:

“The survey was finally sent to all the Heads of Mental Health Departments (MHDs) in the Lombardy Region, Italy, as an initial sample of this research. Thus, the respondents included psychiatrists only, working as Head of Mental Health Departments (MHDs) in the Lombardy Region, Italy, regardless of whether they worked in academic or non-academic settings. No patients, caregivers or other stakeholder completed the survey.”

Moreover, as request by reviewer #3, the Methods section has been carefully revised to improve clarity. Currently, we subdivided the Methods into four sections (2.1 survey construction; 2.2 participants; 2.3 survey aims; 2.4 statistical analyses).

  • The results need to be presented in a way that is clear to the reader. In this current form, they are detailed, but difficult to follow.Please rephrase.

We thank the reviewer #2 for this suggestion. To increase the readability and clarity of the results, we moved some definitions of the “level of consensus”, “degree of implementation” and “gap” from the Results to the Methods. In this way, the definitions are presented earlier, and subsequently the reader can clearly have these concepts in mind when approaching the results.                                                                            

We apologize for repeating each item in full when describing the results since this could create a redundancy for the reader. However, we believe it is important to proceed in this way so that the reader can have a clear idea of the data obtained, analyzing the statements item by item for the three main areas of interest.

Reviewer 3 Report

I commend the authors for this innovative paper. The Patient Journey project aimed to collect experiences on schizophrenia management in clinical practice, focusing on early detection and management, acute phase management, and long-term management/continuity of care. A survey was co-designed with stakeholders and administered to Heads of Mental Health Services in Lombardy Region, Italy. Strong consensus was found for all areas, but implementation was moderate-to-good for early diagnosis and management and slightly above the cut-off for long-term management/continuity of care. The survey highlighted priority intervention areas and current limitations, emphasizing the need for further implementation of early phases and chronicity management to improve the patient journey of schizophrenia patients. 

The major issues of this paper is the lack of clear description of the methods. Specifically I ask authors to: 

Give concise and thorough explanations of the study's sampling plan, data collection techniques, and data analysis procedures. This will make the study's logic more clear to readers and make the findings seem more reliable. How schizophrenia defined ICD10/DSM5. PANSS?

Give readers background on the study's populations (dr, pts, others) and sampling procedures (for each with equations) to help them understand how broadly applicable the results are.

Give specifics on the data collection methods or tools that were used in the study, along with information on their validity and reliability.

Describe how the study dealt with missing data and how any potential biases were reduced.

Clearly and succinctly describe the statistical techniques you used to analyze the data, mentioning any presumptions you made and how you tested them.

Finally, I have concerns that results provide artificial conclusions. All means suggest 4.5 around the score. Thus, either analysis plan was incorrect reliance on mean instead of mode or median. Or the data was not cleaned for outliers. Scoring with this ranges mean that Drs, Pts, and others basically did not know what is important and what is not. 

Author Response

Reviwer #3

We thank the reviewer #3 for the important comments to improve the quality of our work. To improve clarity, we subdivided the Methods into four sections (2.1 survey construction; 2.2 participants; 2.3 survey aims; 2.4 statistical analyses). In this way, we hope to have carefully answered to all the suggestions. We attached the extended revised version of Methods at the end of our answers.

In particular, we hope to have answered to your question “Give concise and thorough explanations of the study's sampling plan, data collection techniques, and data analysis procedures. This will make the study's logic more clear to readers and make the findings seem more reliable. How schizophrenia defined ICD10/DSM5. PANSS?” by having thoroughly changed the Methods section. Thereby, the sampling plan and data collection is addressed in section 2.1, 2.2 and 2.4.

The section 2.1 “survey construction” is aimed to describe the different phases for the construction of the survey and the involvement of all the stakeholders (i.e., clinicians, expert patients, caregivers, and family members) engaged in the planning of the ideal path of care for patients with schizophrenia. Moreover, the section 2.1 “survey construction” is aimed to answer to your question “Give specifics on the data collection methods or tools that were used in the study, along with information on their validity and reliability”. Here, we specify that patients diagnosed with schizophrenia participated only in the shared process with other stakeholders to identify the statements that will compose the survey. They were labeled as “Expert Peer Supporters patients (aka, ESP patients) in the manuscript, but none of these were able to answer to the survey. A definition of ESP patients is now available in the manuscript in section 2.1.

In 2.2 section “participants”, we wanted to answer to your question “Give readers background on the study's populations (dr, pts, others) and sampling procedures (for each with equations) to help them understand how broadly applicable the results are”. In line with reviewer #2 request to clearly state the characteristics of the participants, we described that the participants in this study involved only psychiatrists working as Head of Mental Health Departments (MHDs) in the Lombardy Region, Italy, regardless of whether they worked in academic or non-academic settings. No patients or other individuals completed the survey. Expert patients and caregivers’ associations representatives were only involved in drafting and processing the survey items. To increase clarity, we added the following sentence in the Methods 2.2 section: “no patients, caregivers or other stakeholder completed the survey”.

The section 2.3 “survey aims” was specifically created to make the study more logic and clear for the reader.

According to the suggestion “Describe how the study dealt with missing data and how any potential biases were reduced”, we clearly described in the Results that no missing data was found since all 25 Heads who responded to the survey filled in all statements.

According to the suggestion “Clearly and succinctly describe the statistical techniques you used to analyze the data, mentioning any presumptions you made and how you tested them”, and in line with reviewer #2 requests, we changed and improved the section 2.4 “statistical analyses”.

According to the suggestion “Finally, I have concerns that results provide artificial conclusions. All means suggest 4.5 around the score. Thus, either analysis plan was incorrect reliance on mean instead of mode or median. Or the data was not cleaned for outliers. Scoring with this ranges mean that Drs, Pts, and others basically did not know what is important and what is not”, we believe that results are not exceeding or overrated. Indeed, strong levels of consensus on the “importance of the statement” subscale were quite expected and confirm that the Heads of Lombardy MHDs are aware of the importance of the suggested approaches driven from best practices guidelines and national regulatory sources for the management of schizophrenia. At the same time, high scores on the “degree of implementation” subscale tell us that the Heads of Lombardy MHDs have already implemented the management of schizophrenia in their clinical setting. Moreover, high scores are also expected since the statements of the survey analyzed several clinical practices already widely recognized and learned by the scientific community and applied in clinical practice. However the scenario is not always bright: by analyzing the gaps, we found several limitation of the journey of patients with schizophrenia in the Lombardy MHDs, particularly affecting both the early phases (e.g., the moment of transition from CAMHs to AMHs, the involvement of GPs and family members) and the management of chronicity (e.g., psychological support for family members, provision of semi-residential interventions, the monitoring of patients’ lifestyle and physical health in collaboration with GPs, the integration of experts patients in peer support and multi-professional team). These factors represent real and objective data in the scenario of Lombardy MHDs, as clearly described in the Discussion, representing unmet needs to which the Lombardy MHDs must respond to further improve the patient’s journey of people living with schizophrenia. These factors have been highlighted in the Discussion.

Nevertheless, as suggested by the reviewer, in order to verify if the data obtained could change using mode and median (and not the mean scores), we calculated mode and median for each subscale in the three areas of the survey: we observed that the results did not change substantially as can be seen from the attached table describing the mean, mode and median values. As the reader can see, the standard deviations are low: this data could indicate that the use of mean score could better catch the variability of the responses of the survey. In other word, the range is so small that it's not worth looking for outliers.

Importance of the statement

Early detection

Degree of implementation

Early detection

Importance of the statement

Acute phase

Degree of implementation

Acute phase

Importance of the statement

Long-term management

Degree of implementation

Long-term management

Importance of the statement

Total scores

Degree of implementation

Total scores

Means

4,63

3,97

4,69

4,30

4,62

4,02

4,63

4,07

Median

4,64

3,96

4,72

4,36

4,64

4,16

4,68

4,14

Mode

4,48

3,68

4,72

4,40

4,80

4,16

4,84

4,28

Standard deviations

,224

,410

,173

,279

,258

,448

,226

,415

Round 2

Reviewer 3 Report

thank you for addressing most of my concerns. the methodology section is now very clear.

the results presented in the response letter need to be more transparent and included in the actual paper thus I urge authors to present the median mode and sd in the specific table of the paper. pls check the mode values I expected to see whole integers. 

also please revise your data availability statement to be in more standard format there is no data associated presented in the manuscript (only raw results). 

Author Response

We thank again the Reviewer #3 for taking the time to provide further suggestions to improve the manuscript. Changes in the text are highlighted in green.

1. Request:

The results presented in the response letter need to be more transparent and included in the actual paper thus I urge authors to present the median mode and sd in the specific table of the paper. pls check the mode values I expected to see whole integers. 

Answer:

We added a table (labeled as Table 2 in the manuscript) to provide specific median, mode values and SD and to be more transparent describing our results. We clearly stated in the Statistic section that we also calculated the median and mode values.

We still believe that the use of mean scores could give the possibility to better grasp the variability of the answers of the survey. If we use only the median or the mode values, having to interpret a 5-points Likert scale, we would risk flattening and homogenizing the results, losing the possibility of observing the variability of the data which can instead be obtained from the use of mean scores values. Thus, we added the following sentence in the Statistics section: “For the interpretation of the results, we primary focused on mean scores values”.

2. Request:

Also please revise your data availability statement to be in more standard format there is no data associated presented in the manuscript (only raw results). 

Answer:

We changed the data availability statement as it follows: “The data presented in this study are available on request from the corresponding author”.